# Patient and Family Financial Burden in Cancer: A Focus on Differences across Four Provinces, and Reduced Spending Including Decisions to Forego Care in Canada

Christopher J. Longo [1,2,*], Tuhin Maity [1], Margaret I. Fitch [3] and Jesse T. Young [2,4,5,6,7,8]

1   DeGroote School of Business—Health Policy & Management, McMaster University,
    Hamilton, ON L8S 4L8, Canada; maityt@mcmaster.ca
2   Dalla Lana School of Public Health, University of Toronto, Toronto, ON M5S 1A1, Canada;
    jesse.young@camh.ca
3   Bloomberg Faculty of Nursing, University of Toronto, Toronto, ON M5S 1A1, Canada; marg.i.fitch@gmail.com
4   Institute for Mental Health Policy Research, Centre for Addiction and Mental Health,
    Toronto, ON M5T 1R8, Canada
5   Centre for Epidemiology and Biostatistics, Melbourne School of Population and Global Health,
    The University of Melbourne, Parkville, VIC 3052, Australia
6   Centre for Adolescent Health, Murdoch Children's Research Institute, Parkville, VIC 3052, Australia
7   School of Population and Global Health, The University of Western Australia, Crawley, WA 6009, Australia
8   Curtin School of Population Health, Curtin University, Bentley, WA 6102, Australia
*   Correspondence: cjlongo@mcmaster.ca

**Abstract:** Goal: This study aimed to examine provincial differences in patient spending for cancer care and reductions in household spending including decisions to forego care in Canada. Methods: Nine-hundred and one patients with cancer, from twenty cancer centers across Canada, completed a self-administered questionnaire (P-SAFE version 7.2.4) (344 breast, 183 colorectal, 158 lung, and 216 prostate) measuring direct and indirect costs and spending changes. Results: Provincial variations showed a high mean out-of-pocket cost (OOPC) of CAD 938 (Alberta) and a low of CAD 280 (Manitoba). Differences were influenced by age and income. Income loss was highest for Alberta (CAD 2399) and lowest for Manitoba (CAD 1126). Travel costs were highest for Alberta (CAD 294) and lowest for British Columbia (CAD 67). Parking costs were highest for Ontario (CAD 103) and lowest for Manitoba (CAD 53). A total of 41% of patients reported reducing spending, but this increased to 52% for families earning <CAD 50,000 per year. The highest national rates of decisions to forego care were in relation to vitamins/supplements, the selection made by 21.3% of those who indicated spending reductions. Reductions for complementary and alternative medicine (CAM) were made by 16.3%, and for drugs, by 12.8%. Most cost categories had higher proportions of individuals who decided to forego care when family income was <CAD 50,000 per year and for patients under 65 years of age. Conclusions: Levels of financial burden for patients with cancer in Canada vary provincially, including for OOPC, travel and parking costs, and lost income. Decisions to forego cancer care are highest in relation to vitamins/supplements, CAM, and drugs. Provincial differences suggest that regional health policies and demographics may impact patients' overall financial burden.

**Keywords:** cancer; self-administered questionnaire; out-of-pocket costs; financial burden; financial toxicity; foregone care; regional differences

## 1. Introduction

The most recent published direct medical costs for cancer care in Canada are estimated at CAD 18.4 billion for 2021 [1], including both public and private costs. Additional direct costs included out-of-pocket costs (OOPCs) at CAD 3.1 billion and time costs at CAD 2.0 billion borne by patients and caregivers. Lastly, indirect costs related to lost income were CAD 2.7 billion (all base-case estimates).

Canada's publicly funded health care system suggests that access to health care should not be impeded by financial constraints. However, multiple Canadian studies have suggested that for cancer care, financial burden can exceed CAD 2000 per month when considering out-of-pocket costs and lost income [2–6]. Additionally, a substantial economic burden of OOPCs in cancer has been shown in other publicly funded countries including the UK, Australia, and Ireland [7–10]. Furthermore, although these regions seem to show a lower financial burden, they are still significant. In the US public/private environment, these burdens do tend to be higher than in Canada [11]. Financial burdens are common for patients with cancer, including their families, because not all health care services are fully funded. Co-payments, unfunded care, and lost income for patients and families can result in financial challenges, as shown in both qualitative [12–14] and quantitative studies [6,15]. This financial burden is labeled as financial toxicity and is described as combining "…both objective financial burden and subjective financial distress as key components of financial toxicity (pg 2)" [16]. Policies such as income replacement and means-based medical care programs are often in place to mitigate these financial burdens/toxicities for some patients and their families.

What is much less clear is whether service limits or exclusions differ substantially across provinces for patients with cancer at a financial level, and whether these differences may result in more or less consumption of health services, which could result in sub-optimal care. Hence, we hope to determine if differences in patient spending exist in cancer care, and whether there are differences in the proportion of patients with cancer who decide to forego care.

In Canada, ten provinces and three territories are primarily responsible for health care delivery. The Canada Health Act (CHA) [17] defines terms and conditions which must be adhered to by provinces and territories to receive federal assistance. The CHA specifies the specific health services (i.e., medically required hospital and physician services only) which the federal government agrees to co-fund with the provinces and territories. The CHA came into effect in 1984 at a time when the majority of health services were delivered by physicians or in hospitals. As health care delivery has evolved in Canada and more diverse care delivery models and settings have developed, that historical model of hospital and physician provision does not reflect the current reality. Health care services delivered outside the hospital/beyond the physician, and hence outside the requirements of the CHA, can include items such as outpatient prescription drugs, home care, allied health care, complementary and alternative medicines (CAMs), vitamins/supplements, devices, family support, and other direct treatment-related charges. Consequently, each of the ten provinces and three territories handles health services outside of the CHA quite differently. These differences include the following: 1/different approaches to prescription drugs (annual deductibles, prescription deductibles, and differences in the list of covered drugs); 2/differences in home care services provided including service limits or the types of services provided; 3/differences in travel reimbursement from rural settings, with some provinces offering no assistance; 4/differences in coverage for diagnostic tests.

The existing gaps and challenges related to access to services have been examined in Canada [2–5] and elsewhere [6–10] but few research studies have examined the full extent of the financial burden to patients. It is also uncommon to examine financial burdens across jurisdictions within a country. Only one study with urban versus rural or regional comparisons was identified [18]. The jurisdictional examination may not make sense in countries with centrally administered health care, where variation in coverage is not expected from region to region within a country. However, in countries like Canada, where health care is a provincial responsibility, with variability between provinces in terms of coverage and delivery, an examination across provinces is of interest to understand how provincial policies influence the extent and impact of financial burdens on patients with cancer and their families.

Specifically, there is concern regarding the impact of OOPCs on patients' choices regarding treatment, especially for those in lower income categories. Dunlop and Coyte

(2000) have shown that patients with lower socioeconomic status are less likely to utilize specialist services [19]. Studies of radiation usage patterns in Ontario cancer centers show that patients with lower incomes have a lower chance of receiving radiotherapy to treat their breast cancer [20] or for palliation of cancer symptoms [21]. Although the motivations or conditions driving these income-based differences in utilization are unclear, they may be influenced by OOPCs in cases where patients with limited financial means make decisions to forego required medical treatments or to accept higher health expenditures.

A national study was conducted to 1/quantify financial burdens experienced by individuals diagnosed with cancer in the current Canadian environment; 2/identify differences in relation to patients' perceived burden; 3/compare differences across income, education, and age categories; 4/compare differences across tumor types (breast, colorectal, lung, and prostate); and 5/examine the impact on both patients' and caregivers' incomes [6]. The quantitative results published from this survey document patients' experience related to OOPCs across various cost categories and lost income. The qualitative results reveal various levels of emotional distress related to the financial burden experienced and various strategies used to cope with the situation [13].

In this current analysis, we further analyzed these survey data to examine provincial differences in various cost categories, and additionally explored spending decisions made by patients and families, including decisions to forego cancer care. This analysis reflects different questions from the prior primary analyses (reference) and the findings have yet to be published. The focus of the current study is to present this analysis on provincial variation in OOPCs and spending in greater detail and highlight the policy implications based on our results.

## 2. Patients and Methods

### 2.1. Patient Population

Eligibility for the survey included the following criteria:
1/Patients 18 years of age or older.
2/Ability to read and write in English.
3/One month or longer on treatment, and ideally still on active treatment.
4/Diagnosed with breast, colorectal, lung, or prostate cancer (the most common tumors, representing 48% of all cancers in Canada; https://www.cancer.ca/en/cancer-information/cancer-101/cancer-statistics-at-a-glance/?region=on) (accessed on 5 April 2024).

Recruitment

The study patients (N = 901) were enrolled at 20 cancer centers across Canada (4 in BC, 6 in Alberta, 1 in Saskatchewan, 1 in Manitoba, 6 in Ontario, and 2 in the Atlantic provinces) between 1 May 2016 and 31 May 2019. Centers accrued patients over a 3–18-month period. To allow for a sufficient sample size to compare across provinces, our analysis focuses on the four provinces, British Columbia, Alberta, Manitoba, and Ontario, that had the highest study enrollment and represent 63.5% of the Canadian population (2021 census). Most patients were recruited in person (i.e., non-randomized convenience sample) during cancer clinic visits if they met the eligibility criteria and where primary care providers or research associates (RAs) were available. Some centers enrolled patients through other methods including poster advertising which directed those interested in participating to an online data entry tool for the study, through regular mail using registry data addresses (Manitoba only), and through the use of internet panels (Recruited by Asking Canadians). Cancer clinics were instructed to accrue equal numbers of each tumor type, recognizing that this may not be possible in all centers. Response rates could not be calculated due to the lack of information about a denominator with our recruitment strategies. Additional details on recruitment methods are outlined in our initial publication [4].

*2.2. Data Sources*

The Patient Self-Administered Financial Effects questionnaire (P-SAFE v7.2.4) was used for patients recruited in person and via the internet. The questionnaire was designed to a significant degree based on previous research conducted by Birenbaum [22] and Moore [23] and earlier versions of the P-SAFE questionnaire [2]. A shorter timeframe of 28 days was chosen as longer timeframes have been shown to be unreliable, especially for low ticket cost items [24].

The PSAFE questionnaire included 31 questions with primarily closed-ended formats. Questions covered topics including patient demographics, cancer treatment (e.g., chemotherapy, radiation, and surgery), health care (e.g., doctor visits, emergency room visits, hospitalizations, in-home nursing services, and physiotherapy services), level of insurance coverage, employment details, OOPCs, perceived financial burden, reduced spending including decisions to forego care, and time lost from work for patients and their caregivers. The PSAFE questionnaire did not have information on the stage of disease, as it was suspected that patient-reported disease stages were unreliable. Although we considered requesting access to patient files to determine the stage of disease, it was not pursued due to concerns that it might lead to reduced recruitment.

The PSAFE 'type of expenses' included the following: travel costs, prescription drugs, in-home health care, homemaking services, complementary and alternative medicine (CAM), vitamins and supplements, family care, care by other health professionals, accommodations/meals, devices/equipment, and other costs. Details of the actual questions and examples are included in the Supplementary Materials. This includes questions on foregone care (Q11 Supplementary Materials) and subcategories of medications; home care; homemaking; complementary and alternative therapy; vitamins and supplements; family care; accommodation or meals; devices or equipment; other (specify).

*2.3. Sample Size Calculations*

This study used total OOPCs as the primary outcome to determine the sample size. Consequently, the sample size for secondary outcomes of provincial differences and decisions to forego care may not be adequate to detect statistically significant differences.

*2.4. Calculations and Scoring*

Imputed travel costs were calculated based on travel distance to the clinic, multiplied by the number of trips, and then multiplied by 0.58 CAD/Km based on Canada Revenue Agency mileage rates in 2019, representing the highest enrollment period of study recruitment (https://www.canada.ca/en/revenue-agency/services/tax/businesses/topics/payroll/benefits-allowances/automobile/automobile-motor-vehicle-allowances/automobile-allowance-rates.html) (accessed on 5 April 2024). The data from patients and caregivers who lost time from work in the previous 28 days were used together with family income data to determine crude estimates of income lost. 'Partial pay' was indicated by some patients and an actual number or percentage loss in dollars was not provided. In this regard, our estimates are likely high.

Where required, determination of average family income for each participant was achieved by calculating the midpoint of the "family income" category that participants had chosen in the questionnaire, with the value for those earning > CAD 100,000/year entered as CAD 110,000, a conservative estimate.

In this analysis, there were sufficient subjects to examine differences between the provinces of British Columbia, Alberta, Manitoba, and Ontario. Although we accrued an additional 51 patients across the remaining 6 provinces, the relatively small samples made these provincial comparisons unreliable.

Decisions by patients to forego care or other basic needs was examined and analyzed across several factors/subgroups (Q11 in Supplementary Materials). This included looking at all patients together and patients belonging to two subgroups: patients with a family income <CAD 50,000 and those patients below the age of 65, considered higher risk groups.

The questions on 'decisions to forego care' were presented in two parts. The first asked if participants incurred spending changes due to financial barriers associated with their cancer treatments. Secondarily, we asked about the nature of these changes in spending. In most cases, more than 95% of the data fields were complete. The data were presented as the percentage of patients who decided to forego care across the full population and, secondarily, only for those who made reductions in spending.

### 2.4.1. Descriptive Statistics

Information on patient demographics and tumor type was captured and is presented as means, standard deviations, and ranges.

### 2.4.2. Analyses of Variance (ANOVAs)

ANOVAs were performed to identify differences in the dependent variable between different groups (independent variables) including province and income categories. In cases where ANOVAs showed statistically significant differences, the Tukey Honest(ly) Significant Difference (HSD) post-hoc test [25] was used to identify group differences. In instances where t-tests were undertaken, we applied Levene's test for equality of variances to determine if the variances were equal and the appropriate t-test variance specification was then applied.

### 2.4.3. Chi-Square

A chi-square test was employed to determine differences between groups. This included comparisons of differences across age categories and income categories related to decisions to forego care. Secondarily, we examined differences in comparison to Canadian Cancer Registry (CCR) data (2018–2019) to determine if our sample was generalizable to the Canadian population.

### 2.5. Multivariable Linear Regressions

We undertook an examination of total out-of-pocket costs, set as the dependent variable, and included the following independent variables: province (4), age (continuous), income category (<CAD 50,000 vs. >CAD 50,000), and patient burden category (high vs. low).

All analyses were performed using the statistical software RStudio v1.2.1335 [2019], based on R platform v3.6.

### 2.6. Ethics

Ethics approval was obtained from McMaster University's Hamilton Integrated Research Ethics Board (HiREB #1743). Additionally, site ethics approvals were obtained from each of the local cancer centers.

## 3. Results

### 3.1. Participants

The patients (N = 901) were evenly divided between male and female, with 482 out of the total 896 (53.8%) being females (5 patients declared gender "other"). The male/female mix varied by tumor type (Table 1). There were several instances where patients chose not to answer particular questions and hence the full 901 patients' responses were not available in all data fields (e.g., 8.2% selected "Don't know/missing" for income). The questionnaire took 20–35 min to complete online. Clinic staff did not track the time it took to complete the in-person questionnaire, although it appeared to be faster than online as no delays due to screen loading existed in person.

**Table 1.** Study population: demographic characteristics by study sample and tumor type.

|  |  | Total (n = 901) | Breast (n = 344) | Colorectal (n = 183) | Lung (n = 158) | Prostate (n = 216) |
|---|---|---|---|---|---|---|
|  | Age | 61.3 | 55.5 | 59.0 | 65.3 | 67.6 |
|  | Age Range | 20–92 | 20–84 | 25–92 | 20–89 | 23–90 |
|  | Male | 414 | 6 | 113 | 81 | 214 |
|  | Female | 481 | 336 | 70 | 75 | 0 |
|  | Treatment Duration | 318 d | 307 d | 320 d | 339 d | 318 d |
| Education |  | n (%) | n (%) | n (%) | n (%) | n (%) |
|  | Elementary | 19 (2.1) | 2 (0.6) | 3 (1.6) | 8 (5.1) | 6 (2.8) |
|  | Some HS | 55 (6.1) | 10 (2.9) | 15 (8.2) | 16 (10.1) | 14 (6.5) |
|  | Compl HS | 169 (18.8) | 68 (19.8) | 39 (21.3) | 32 (20.2) | 30 (13.9) |
|  | Some Univ | 170 (18.9) | 58 (16.9) | 30 (16.4) | 38 (24.1) | 44 (20.4) |
|  | Compl Univ | 367 (40.8) | 161 (46.9) | 75 (40.1) | 52 (32.9) | 79 (36.6) |
|  | Post Grad | 120 (13.3) | 44 (12.8) | 21 (11.5) | 12 (7.6) | 43 (19.9) |
| Income |  | n (%) | n (%) | n (%) | n (%) | n (%) |
|  | CAD 0–19.9 K | 65 (7.2) | 19 (5.5) | 17 (9.3) | 21 (13.3) | 8 (3.7) |
|  | CAD 20–39.9 K | 137 (15.2) | 50 (14.6) | 33 (18.0) | 29 (18.4) | 25 (11.6) |
|  | CAD 40–59.9 K | 142 (15.8) | 54 (15.7) | 27 (14.8) | 28 (17.7) | 33 (15.3) |
|  | CAD 60–79.9 K | 134 (14.9) | 52 (15.2) | 28 (15.3) | 24 (13.0) | 30 (13.9) |
|  | CAD 80–99.9 K | 131 (14.6) | 52 (15.2) | 22 (12.0) | 13 (8.2) | 44 (20.4) |
|  | CAD 100 K plus | 218 (24.2) | 90 (26.2) | 42 (23.0) | 25 (15.8) | 61 (28.2) |
|  | Missing/DK | 73 (8.1) | 26 (7.6) | 14 (7.7) | 18 (11.4) | 15 (6.9) |

Note 1: five patients declared gender as "other". Note 2: all costs are in Canadian dollars.

An analysis comparing primary outcome (out-of-pocket costs) did not show any statistically significant differences across methods of data capture (e.g., paper vs. electronic).

The mean age of the patients in the full sample was 61.3 years. There was a noticeable degree of variability in age between tumor types. Patients with breast cancer were the youngest (mean = 55.5 years) and patients with prostate cancer were the oldest (mean = 67.6 years) (Table 1).

The patients had a somewhat skewed education distribution, with 8.2% of the sample having less than a high school education and 73.0% having at least some university/college exposure (Table 1). The average duration of treatment for the patients was just under 1 year, at 318 days, and this ranged from 25 days to 2.7 years.

### 3.2. Generalizability of the Study Population

In comparison to Canadian Cancer Registry data across gender and age, our results showed that our sample was similar to all cancer patients in terms of gender balance for both lung ($p = 0.367$) and colorectal cancer ($p = 0.667$). It also aligned in terms of age for prostate cancer ($p = 0.9421$). However, our sample was significantly younger for breast ($p < 0.001$), colorectal ($p < 0.001$), and lung cancers ($p = 0.001$), when compared to the Canadian Registry population.

#### 3.2.1. National Patient Expenditures

Aggregate mean monthly OOPC for all patients with cancer was CAD 518, over 28 days, with an additional CAD 179 related to imputed travel, CAD 84 for parking, and a combined CAD 1733 for patient and caregiver lost income (Table 2). The resulting overall cost for the 28 days was CAD 2514 nationally.

**Table 2.** National and provincial 28-day expenditures for OOPC, parking, travel, and lost income versus mean monthly family income.

|  | National (n = 901) | BC (n = 131) | Alberta (n = 113) | Manitoba (n = 134) | Ontario (n = 472) |
|---|---|---|---|---|---|
| OOPC | **CAD 518** | CAD 528 | CAD 938 | CAD 280 | CAD 461 |
| Parking | **CAD 84** | CAD 62 | CAD 71 | CAD 53 | CAD 103 |
| Travel | **CAD 179** | CAD 67 | CAD294 | CAD 146 | CAD 194 |
| Lost income Pt and caregiver [1] (days) | **CAD 1733 (8.3)** | CAD 1668 (8.0) | CAD 2399 (11.5) | CAD 1126 (5.4) | CAD 2010 (9.6) |

**Table 2.** *Cont.*

| | National (n = 901) | BC (n = 131) | Alberta (n = 113) | Manitoba (n = 134) | Ontario (n = 472) |
|---|---|---|---|---|---|
| Total financial effect | **CAD 2514** | CAD 2325 | CAD 3702 | CAD 1605 | CAD 2768 |
| Family income (percentage) [2] | **CAD 5304 (47.4)** | CAD 5211 (44.6) | CAD 5539 (66.8) | CAD 4562 (35.2) | CAD 5522 (50.1) |

[1] The cost estimate for lost income is based on the days lost multiplied by the national average daily income in Canadian dollars. [2] The percentage of family income was used to provide a sense of what proportion of income in that 28-day period would be needed to cover these cancer-related costs.

### 3.2.2. Provincial Differences in OOPC

The differences in total OOPCs, parking, imputed travel costs, and lost time from work for patients and their caregivers across these four provinces are presented in Table 2. Alberta had the highest overall costs and Manitoba had the lowest overall costs (CAD 3702 vs. CAD 1605 per 28 days). However, BC had the lowest 28-day travel costs (CAD 67 vs. a high of CAD 294 in Alberta). These provincial differences were mainly driven by higher OOPCs and greater losses of patient and caregiver income. An ANOVA showed a statistically significant difference across provinces for OOPCs ($p = 0.003$), with differences occurring between Ontario and Alberta ($p = 0.009$) and between Manitoba and Alberta ($p = 0.002$). Additionally, an ANOVA for 28-day family income across provinces showed a statistically significant difference ($p = 0.014$), with statistically significant differences only occurring between Manitoba and Ontario ($p = 0.010$).

The proportion of patients declaring a high burden (somewhat, large, or worst) varied by province, with a high of 36.7% in Ontario and a low of 25.4% in Manitoba. A chi-square test comparing the provinces was statistically significant ($p = 0.0474$).

### 3.2.3. Provincial Differences in Time Lost from Work for Patients and Caregivers

Of those patients who had worked in the past 28 days, the mean lost time from work was 18.0 days (5.3 days across the entire sample). Moreover, many caregivers who worked also lost time; the 26% that took time off work in relation to cancer care averaged 11.5 days off (an average of 3.0 days off for the entire sample).

Across all subjects, the average number of lost days of work for patients and their caregivers varied between provinces. The raw number of days lost were 8.0 days in BC, 11.5 days in Alberta, 5.4 days in Manitoba, and 9.6 days in Ontario. Nationally, the average was 8.3 days lost from work in the previous 28 days for patients and their caregivers.

### 3.2.4. Provincial Differences for Travel and Parking

British Columbia had the lowest 28-day travel costs (CAD 67), while Alberta had the highest (CAD 294). This is partly reflected in the average travel distance to health services, with distances per 28 days of 507 km for Alberta, 115 km for British Columbia, 251 km for Manitoba, and 335 km for Ontario. Twenty-eight-day parking costs were lowest for Manitoba (CAD 53) and highest for Ontario (CAD 103) [Table 2].

### 3.2.5. Decisions to Forego Care

An examination of decisions by patients to forego care based on financial grounds was conducted. The data are presented looking at all patients together and then at two subgroups: patients with an income below CAD 50 K and those below the age of 65 (higher risk groups) [Table 3]. Overall, 40.7% of the sample made changes to their spending due to financial issues related to their cancer. This increased to 52.2% for those making less than CAD 50,000 per year versus 37.0% for those earning over CAD 50,000 (chi-square $p < 0.001$), and to 54.3% for those under the age of 65 versus 24.2% for those over 65 (chi-square $p < 0.001$). In looking specifically at health care decisions, the greatest effect was seen when examining drugs and vitamins/supplements in the sample earning less than CAD 50,000, with 20.7% of those foregoing medications and 29.0% foregoing vitamins/supplements. These numbers were only slightly lower for those under 65, with 13.4% foregoing drugs

and 21.9% forgoing vitamins/supplements. Other cost categories showed lower rates of foregone care.

**Table 3.** Decisions to forego care as percent of full sample (ALL), percent with lower income (Income < CAD 50 K), and percent for those under 65 years of age (Age < 65).

|  | National (N = 901) [1] | BC (N = 131) | Alberta (N = 113) | Manitoba (N = 134) | Ontario (N = 472) |
|---|---|---|---|---|---|
| **Adjust Spend (ALL)** | **40.7** | **39.7** | **36.3** | **35.1** | **42.8** |
| Income < CAD 50 K | 52.2 | 42.6 | 46.2 | 52.7 | 57.9 |
| Age < 65 | 54.3 | 55.9 | 47.1 | 58.3 | 55.1 |
| **Drugs (ALL)** | **12.8 (5.2)** | **13.5 (5.3)** | **12.2 (4.4)** | **14.9 (5.2)** | **10.9 (4.7)** |
| Income < CAD 50 K | 20.7 (10.8) | 25 (10.6) | 0 (0) | 20.7 (10.9) | 19.5 (11.3) |
| Age < 65 | 13.4 (7.3) | 15.8 (8.8) | 12.1 (5.7) | 21.4 (12.5) | 10.7 (5.9) |
| **Devices (ALL)** | **9.0 (3.7)** | **7.7 (3.1)** | **17.1 (6.2)** | **8.5 (3)** | **6.9 (3)** |
| Income < CAD 50 K | 8.3 (4.3) | 5.0 (2.1) | 8.3 (3.8) | 10.3 (5.5) | 7.8 (4.5) |
| Age < 65 | 9.3 (5.1) | 10.5 (5.9) | 21.2 (10) | 10.7 (6.3) | 5.3 (2.9) |
| **Home care (ALL)** | **8.7 (3.6)** | **9.6 (3.8)** | **14.6 (5.3)** | **8.5 (3)** | **7.4 (3.2)** |
| Income < CAD 50 K | 10.3 (5.3) | 15 (6.4) | 0 (0) | 13.8 (7.3) | 9.1 (5.3) |
| Age < 65 | 8.6 (4.6) | 13.2 (7.4) | 15.2 (7.1) | 7.1 (4.2) | 6.7 (3.7) |
| **CAM (ALL)** | **16.3 (6.7)** | **17.3 (6.9)** | **24.4 (8.8)** | **14.9 (5.2)** | **14.9 (6.4)** |
| Income < CAD 50 K | 14.5 (7.6) | 10 (4.3) | 8.3 (3.8) | 17.2 (9.1) | 15.6 (9.0) |
| Age < 65 | 19.3 (10.5) | 21.1 (11.8) | 27.3 (12.9) | 21.4 (12.5) | 16.8 (9.2) |
| **Vitamins (ALL)** | **21.3 (8.7)** | **21.2 (8.4)** | **24.4 (8.8)** | **23.4 (8.2)** | **20.3 (8.7)** |
| Income < CAD 50 K | 29 (15.1) | 25 (10.6) | 8.3 (3.8) | 31 (16.4) | 29.9 (17.3) |
| Age < 65 | 21.9 (11.9) | 23.7 (13.2) | 27.3 (12.9) | 28.6 (16.7) | 20.1 (11.1) |

[1] National patient total of 901 includes 51 patients from Quebec, Saskatchewan, and Atlantic Canada combined. Note A: in the subcategories, the main number for Drugs, Devices, Home care, CAM, and Vitamins is the percentage of those that declared changes in spending vs. the percentage in the full sample including those with no spending changes. Note B: CAM = complementary and alternative medicine. Note C: all incomes are reported in Canadian dollars.

### 3.3. Multivariable Linear Regression

Lastly, we ran a linear regression model with total out-of-pocket costs as the dependent variable, and provinces (four), income category (<CAD 50,000 vs. ≥CAD 50,000), age (continuous), and patient burden (high vs. low) as the independent variables. Our results showed that Alberta had significantly higher costs than BC, Manitoba, and Ontario, low income significantly reduced spending, increased age significantly reduced costs, and high patient burden significantly increased spending (Table 4).

**Table 4.** Multivariable linear regression of out-of-pocket costs by province (4), income category (≥CAD 50 K vs. <CAD 50 K), age, and burden category (high vs. low).

|  | **Dependent Variable** |  | **Out-of-Pocket Costs (OOPCs)** |  |
|---|---|---|---|---|
| *Variable* | *Coefficient (SE)* | *Beta* | *95% Confidence Interval* | *p Value* |
| **Province** |  |  |  |  |
| BC | −CAD 399.75 (CAD 195.02) | 2.05 | −CAD 782.18, −CAD 17.32 | $p < 0.05$ |
| Manitoba | −CAD 507.20 (CAD 200.33) | 2.53 | −CAD 900.05, −CAD 114.35 | $p < 0.05$ |
| Ontario | −CAD 532.52 (CAD 160.29) | 3.32 | −CAD 218.19, −CAD 846.85 | $p < 0.01$ |
| **Income** |  |  |  |  |

**Table 4.** *Cont.*

| Variable | Coefficient (SE) | Beta | 95% Confidence Interval | p Value |
|---|---|---|---|---|
| | **Dependent Variable** | | **Out-of-Pocket Costs (OOPCs)** | |
| Below CAD 50 K | −CAD 326.51 (CAD 113.41) | 2.88 | −CAD 548.91, −CAD 104.11 | $p < 0.01$ |
| **Age** | | | | |
| Continuous | −CAD 9.08 (CAD 3.85) | 2.35 | −CAD 16.63, −CAD 1.53 | $p < 0.05$ |
| **Burden** | | | | |
| Burden High | CAD 671.10 (CAD 115.85) | 5.79 | CAD 443.92, CAD 898.28 | $p < 0.01$ |
| Constant | CAD 1329.34 (CAD 270.79) | 4.91 | CAD 798.32, CAD 1860.36 | $p < 0.01$ |

Observations → 772, R2 → 0.089, Adjusted R2 → 0.082, Residual Std. Error → 1442.466 (df = 765), and F Statistic → 12.476 (df = 6; 765). "CAD 50,000+" is income reference, "small burden" is burden reference, and "Alberta" is the province reference. *Hypotheses*: lower OOPCs with increasing age; higher-income patients have higher OOPCs; higher-burden patients have higher OOPCs. NOTE A: standardized beta = coefficient/SE; for 78 pts, income was "don't know" (n = 73) or "missing" (n = 5); 51 pts in other provinces. NOTE B: all costs are reported in Canadian dollars.

## 4. Discussion

The primary goal of this study was to identify patient and family burdens related to cancer treatment and how they might differ across provinces. In addition, decisions to forego care were also examined. This research suggests that provincial differences exist and are statistically significant in some cases. We note that this also translates or is potentially related to decisions to forego care, particularly for those with family incomes below CAD 50,000 per year and those under 65 years of age. We acknowledge that this may or may not be relevant in different settings such as those in other countries and where health care systems may differ.

As noted in a previous Canadian publication [6], about one-third of those patients with cancer found the burden of OOPCs to be "somewhat, large or the worst possible" burden despite having a publicly funded health care environment. The patients in this analysis reported spending an average of 34% of their monthly income on cancer-related costs. It can be speculated that both direct and indirect cost elements play a role in patients' perceived burden, as other cancer research has provided evidence that both cost elements contribute to patient costs [26–28]. These results show that expenditure as a percentage of income is greatest for those with low incomes, consistent with other cancer research [29]. Finally, the limited literature that does exist on OOPCs of patients with cancer is mostly from a predominately private, for-profit health care environment. What has been added to the body of knowledge from this analysis is the notion of patients with cancer making decisions to forego care in an attempt to minimize financial burden or financial toxicity. The findings indicate that this approach to managing financial challenges is relatively common among patients.

Family incomes below CAD 50,000 per year increase the likelihood of deciding to forego care, to report an increased financial burden, and to take time off work. Unfortunately, more time off work exacerbates the financial burden for these lower-income families. Each of these impacts highlights the particularly problematic issues that arise in these Canadian populations. Although our sample had an under-representation of this low-income population segment, we had sufficient numbers to identify statistically significant differences across income groups.

In looking at total OOPCs, travel costs, and parking costs, we observed some differences across the four provinces with sufficient sample sizes. Some of that difference may

be influenced by differences in age and income distributions between the provinces as well as buying power in different jurisdictions, although it is unlikely to be the only reason for the differences. Earlier work has suggested that both age and income factors influence patient costs [3]. Our multivariable linear regression model supports the notion that older age results in reduced costs and that low income reduces costs but also increases the risk of foregoing care.

It was noted that a few studies examined foregone care in Atlantic Canada [30,31], but otherwise, no historical data on decisions to forego care in Canadian settings were identified. The frequency of decisions to forego care was not trivial (in some cases approaching 30%, Table 3), especially so for those families earning less than CAD 50,000 per year and patients under 65 years old (Table 3). Although this result is surprising, the numbers observed approach those reported in at least one US study [32]. Local presentations of this research at cancer centers in Canada suggest that some oncologists/social workers were not surprised by these data, suggesting this may be an ongoing issue in Canada in many provinces. These results provide a better sense of the frequency and nature of these patient decisions across provinces.

### 4.1. Limitations and Future Research

This sample was taken from twenty cancer clinics and included the four most common tumor types in Canada. The methods did not include an evaluation of patients treated outside cancer clinics, patients with other cancers, or patients with cancer who chose not to be treated, although the untreated patients' costs would be nominal. Hence, it cannot be determined if costs would have been higher or lower, and whether those costs would be in the same categories as was seen in this study. Data regarding other financial factors that can impact patients' available disposable incomes, including loans, education savings for family members, and personal saving were not captured, only the intent to use them for cancer-related care. Also, due to resource limitations, we were unable to obtain details on those patients who declined participation or the overall population available for the specific tumor type during the period of analysis. Our comparison to the Canadian Cancer Registry suggests our sample is comparable on gender for colorectal and lung cancer, and on age for prostate cancer. However, our sample was younger in breast, colorectal, and lung cancers which is likely to result in a slight overstatement of costs as older age reduces the patients' out-of-pocket costs. We recognize that the younger age in three of four tumors studied makes assumptions about generalizability limited.

Additionally, when comparing between provinces, we note that buying power between provinces or even between cities is likely different. Hence, a CAD 300 expense in Winnipeg, Manitoba, might be equal to a burden of CAD 600 in Toronto, Ontario, but these types of adjustments were not possible to make with the data available from the PSAFE questionnaire. Income loss calculations are based on actual days lost from work. There are several ways to convert this estimate to dollars, and each approach has limitations in its assumptions. As with previous publications, the national average daily salary (CAD 209 in 2019) was used in an effort to reduce bias between provinces. The actual average individual income after tax in 2019 was CAD 51,200 (cad 211/day) in Ontario; in Manitoba, it was CAD 47,300 (CAD 195/day); in Alberta, it was CAD 58,700 (CAD 242/day); in British Columbia, it was CAD 51,400 (CAD 212/day) [https://www150.statcan.gc.ca/t1/tbl1/en/tv.action?pid=11100239 01&pickMembers[0]=1.1&pickMembers[1]=2.1&pickMembers[2]=3.1&pickMembers[3]=4 .1&cubeTimeFrame.startYear=2017&cubeTimeFrame.endYear=2021&referencePeriods=20 170101,20210101] (accessed on 5 April 2024). In a future publication by the lead author, an examination of several different model assumptions for lost income will be evaluated. It is expected that the modeling of the lost income analysis will introduce a variance of approximately 20–25%, so these income loss costs (stated separately in Table 2) should be interpreted as preliminary.

Since the primary outcome of this research was aggregate OOPCs, sample size calculations were not determined based on the secondary outcomes of provincial differences and

decisions to forego care. Hence, it is possible that some non-significant findings related to these secondary variables may be due to insufficient sample size.

In the case of decisions to forego care, we found our sample sizes to be too small to obtain a robust multivariable regression calculation. Although this would prove illuminating if completed, it was not feasible in this analysis.

It was noted that a high degree of variability exists with the cost for a day of parking both between provinces and between cities within provinces. They ranged from a low of CAD 10 to a high of over CAD 30. In most cases, the high costs were in urban centers like Vancouver and Toronto, with more rural centers being less expensive. Additionally, in urban centers, more individuals will use public transit to avoid the higher parking fees. As a result, comparison between provinces is more challenging to interpret.

It appears that most cancer treatments that require aggressive chemotherapy would, by definition, require a significant expense related to prescription drugs (especially for oral anti-emetics and oral chemotherapy agents). In this regard, it is expected that the gap identified in this research would also be found for other tumor types. Demographics can also play a role, as some cancers tend to have a younger population and hence are more likely to have uninsured or underinsured individuals.

The fact that we did not capture information on the stage of disease means that we are unable to confirm earlier research on whether it significantly influences expenditures. The literature has shown that costs related to care for lung, colorectal, breast, prostate, and bladder patients with cancer tend to be greatest in the first 6 months following diagnosis and in the last year before death, with the time in between significantly less expensive [33]. Typically, patients are recruited in this study throughout the life cycle of their illness; hence, significant variability in health care resource consumption can be expected and may have a significant impact on their OOPCs as well as their perceived financial burden. Clearly, some costs, being episodic in nature, occur early in a patients' treatment, while others occur later in their treatment. These issues make it more difficult to determine the factors that determine those patients most at risk of significant financial burdens.

*4.2. Policy Implications*

In Canada, a number of programs are designed to assist patients with high financial burdens related to health. They include special means-tested drug funding programs, age-related programs, health care funding programs for patients with work-related illnesses, and special health care funding for persons who are out of work. Whether those patients with high financial burden were eligible for and/or aware of such assistance is understudied and requires further investigation. The new information here on decisions to forego care raises alarms that patient health can be compromised due to financial issues. Our analysis of provincial differences highlights the effect of different policy strategies to mitigate these expenses and warrants further study.

One factor that affects perceived burden is related to psycho-social distress, which has been shown to be as high as 43% in patients with lung cancer [34]. This observation suggests there may be an opportunity to minimize these perceived patient burdens through more extensive supportive care initiatives like financial counselling, whether in home or institution settings.

The loss of approximately half of the caregiver's income while providing necessary services or assistance to patients at home contributes greatly to the patients' and families' perceived financial burden. First instituted in December 2017, the Federal Minister of Health provided grants to cover supportive leave for families delivering end-of-life care up to a maximum of 15 weeks for critically ill patients and a maximum of 26 weeks for end-of-life support. This financial support program is presently still in place (August 2023) and will address the needs of many patients with advanced disease. Even so, maximum payments by the government are set at 55% of the full income, which may still create a financial burden for patients and their families. Moreover, the cancers studied in this analysis have six- to twelve-month treatment cycles and often have follow-up treatment

that will carry on significantly longer than these time periods. These time periods would be well beyond the limits of these federal programs.

The results from this analysis suggest that the government limits on health care services can impact or influence other social programs such as those associated with income replacement. It also suggests that differences between provinces could be influenced, outside the Canada Health Act, by policy decisions on the scope of care within provinces. It also raises the question of whether policy makers should look carefully at successful programs outside of their home province as well as outside of the Ministry of Health when evaluating the comprehensiveness of publicly funded health care programs for cancer. As has been shown in other studies, financial burden is particularly problematic for those in lower income categories, and often for those who have not yet reached 65 years of age, when many health-related programs become available through the public purse. This study answers a number of questions about the size and frequency of financial burdens for cancer patients in Canada and among those living in four provinces, generating more opportunities for research. The key findings should be useful to provincial policy makers and allow for a closer evaluation of existing provincial programs in light of the success or failure of programs to mitigate patient cancer-related costs. Additionally, the more serious decision by patients and their families to reduce recommended care due to financial constraints may be improved through policy changes. Whether these strategies would be useful in other countries is difficult to assess based on differences in the mix of public and private health care delivery and the overall structure of health care funding in other jurisdictions.

**Supplementary Materials:** The following are available online at https://www.mdpi.com/article/10.3390/curroncol31050206/s1, Supplementary Materials, Patient Self-Administered Financial Effects (P-SAFE) questionnaire.

**Author Contributions:** Conceptualization, C.J.L. and M.I.F.; methodology, C.J.L. and J.T.Y.; software, C.J.L.; validation, C.J.L., T.M., J.T.Y. and M.I.F.; formal analysis, C.J.L.; investigation, C.J.L. and M.I.F.; resources, C.J.L.; data curation, C.J.L.; writing—original draft preparation, C.J.L.; writing—review and editing, C.J.L., T.M., M.I.F. and J.T.Y.; visualization, C.J.L., T.M., M.I.F. and J.T.Y.; supervision, C.J.L.; project administration, C.J.L.; funding acquisition, C.J.L. All authors have read and agreed to the published version of the manuscript.

**Funding:** Canadian Centre for Applied Research (funded through the Canadian Cancer Society Research Institute, Grant # 2015-703559, year 2015) $185,000; Ontario MoH/Cancer Care Ontario—Planning and Regional programs, Grant # NA, year 2018) $36,000; McMaster Arts Research Board—SSHRC, Grant # NA year 2019) $9000.

**Institutional Review Board Statement:** Ethics approval was obtained from McMaster University's Hamilton Integrated Research Ethics Board (HiREB #1743). Additionally, site ethics approvals were obtained from each of the local cancer centers.

**Informed Consent Statement:** Informed consent was obtained from all subjects involved in this study.

**Data Availability Statement:** The dataset is available upon request from the authors.

**Acknowledgments:** Funding was provided through the Canadian Centre for Applied Research in Cancer Control, Ontario Ministry of Health, and the McMaster Arts Research Board. There were a number of individuals who assisted at the participating regional cancer centers, providing instrumental support for patient recruitment and local study management, including Justin Jao (BC); Emma Tolsdorf (Alb); Zeb Aurangzeb, Elizabeth Harland, and Carrie O'Conaill (Man); Melissa Korman, Laura Goldberg, Anne Malpage, Albert Gratton, Guilio Didiodato, Jesse Maclean, Christine DiMarco, the Royal Victoria Regional Health Centre V-forces team, and Carla Girolametto (Ont); Margaret Jorgensen (NS); and Dana Ryan (Nfld). The clinics included Toronto Sunnybrook Regional Cancer Centre, Juravinski Cancer Center, Princess Margaret Hospital, Grand River Regional Cancer Centre, Simcoe Muskoka Regional Cancer Centre, the London Regional Cancer Program, four BC cancer centers, six Alberta cancer centers (excluding satellite centers), Saskatoon Cancer centre (Saskatchewan), CancerCare Manitoba—McDermot site, Nova Scotia Cancer Centre, and H. Bliss Murphy Cancer Centre (Newfoundland).

**Conflicts of Interest:** CJL, MIF, and TM have none to declare. JTY receives research support from a National Health and Medical Research Council Investigator Grant (GNT1178027).

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
