# Peer review of "Patient and Family Financial Burden in Cancer: A Focus on Differences across Four Provinces, and Reduced Spending Including Decisions to Forego Care in Canada"

_curroncol, doi:10.3390/curroncol31050206_

Round 1

Reviewer 1 Report (Previous Reviewer 1)

Comments and Suggestions for Authors

All comments have been adequately addressed in the revised manuscript.

Comments on the Quality of English Language

Overall no significant concerns with language.

Author Response

We thank reviewer #1 for their previous questions and feedback on the current draft.

Reviewer 2 Report (Previous Reviewer 2)

Comments and Suggestions for Authors

Thank you for the opportunity to review a revised version of this paper. Unfortunately, the authors have not sufficiently responded to my original comments in the revised version. In particular, I do not believe they adequately addressed the significant threats to generalizability associated with the sampling approaches taken at the different cancer centers, nor have they addressed the issues associated with being able to properly characterize the respondents versus non-respondents to the survey. While I understand the limits of small budgets, being able to give a sense as to who the survey respondents are as a proportion of the total potential respondents is a key part of survey research design and should be reported as part of any paper that wishes to convince readers as to its validity.

In addition, the Introduction could use more context regarding health insurance coverage in Canada and how it varies across the provinces for cancer care. Right now there is little context for the reader to understand how much cancer patients might expect to incur OOP for their care.

Further, I do not understand the new paragraph in the Methods section entitled "Sample size calculations", though I have read it a few times. It is not clear to me how the OOP costs were related to the sample size.

The methods section does not provide much information regarding how patients were asked about their OOP costs and other costs associated with cancer treatment. Were they asked to estimate their monthly costs right in the moment for each category? Were the authors concerned about recall bias at all? Were they able to verify OOP costs using claims data or some other administrative dataset?

In Table 2, I'm not sure why a percent is shown after family income? Is this supposed to show the percent of family income lost in a given month? If so, I would recommend better labeling and descriptions of these different outcome measures. Also, why were 28-day costs shown as opposed to the total costs for a course of cancer treatment? The 28 day numbers look very low but I think would be more meaningful if people understood the total costs incurred. Do Canadian cancer patients hit an OOP maximum, similar to what exists in the United States under some insurance schemes? That might limit the overall liability for patients.

Comments on the Quality of English Language

As noted in the previous review, the paper would benefit from review by a copy-editor. The added text in tracked changes contains many errors and it is difficult to follow in many cases.

Author Response

1. Thank you for the opportunity to review a revised version of this paper. Unfortunately, the authors have not sufficiently responded to my original comments in the revised version. In particular, I do not believe they adequately addressed the significant threats to generalizability associated with the sampling approaches taken at the different cancer centers, nor have they addressed the issues associated with being able to properly characterize the respondents versus non-respondents to the survey. While I understand the limits of small budgets, being able to give a sense as to who the survey respondents are as a proportion of the total potential respondents is a key part of survey research design and should be reported as part of any paper that wishes to convince readers as to its validity.

RESPONSE

Thank you for your comments. As a pseudo-comparison we have identified registry data from 2018-2019 to identify how study patients’ demographics differs from the general population of cancer patients. We have added this new information into both the methods and results. Our findings show that we have an appropriate representation by gender, but that our sample is statistically younger in 3 of 4 tumour types. This will slightly overstate the patient burden, as those under 65 have higher costs on average, as many health subsidy and financial support programs begin at age 65. We have revised the results and discussion sections to reflect this.

---------------------------------------------------------------------

2. In addition, the Introduction could use more context regarding health insurance coverage in Canada and how it varies across the provinces for cancer care. Right now there is little context for the reader to understand how much cancer patients might expect to incur OOP for their care.

RESPONSE

Thank you for your comment. More detail has been provided in the Introduction, although we are now over the word count limit.

---------------------------------------------------------------------

3. Further, I do not understand the new paragraph in the Methods section entitled "Sample size calculations", though I have read it a few times. It is not clear to me how the OOP costs were related to the sample size.

RESPONSE

Thank you for your comment. We apologize if this was not clear. The primary outcome of this study was the aggregate out of pocket costs. Examination of provincial differences and foregone care were secondary analyses, thus, no statistical sample size calculations were conducted. We have reworded this section and softened our language to reflect the fact that these were secondary analyses. 

---------------------------------------------------------------------

4. The methods section does not provide much information regarding how patients were asked about their OOP costs and other costs associated with cancer treatment. Were they asked to estimate their monthly costs right in the moment for each category? Were the authors concerned about recall bias at all? Were they able to verify OOP costs using claims data or some other administrative dataset?

RESPONSE

Thank you for your comment. Much of this was addressed in the 2021 publication (Longo, C. J.; Fitch, M. I.; Loree, et al. Patient and Family Financial Burden Associated with Cancer Treatment in Canada: A National Study. Supportive Care in Cancer 2021, 29 (6), 3377–3386. https://doi.org/10.1007/s00520-020-05907-x ). Given these methods have been published previously we have opted to be more concise here. However, we have now added a reference to the original paper and clearly pointed the reader to this reference that provides further details about the methods of our study.

---------------------------------------------------------------------

5. In Table 2, I'm not sure why a percent is shown after family income? Is this supposed to show the percent of family income lost in a given month? If so, I would recommend better labeling and descriptions of these different outcome measures. Also, why were 28-day costs shown as opposed to the total costs for a course of cancer treatment? The 28-day numbers look very low but I think would be more meaningful if people understood the total costs incurred. Do Canadian cancer patients hit an OOP maximum, similar to what exists in the United States under some insurance schemes? That might limit the overall liability for patients.

RESPONSE

Thank you for your comment. We have added some additional information in the table to clarify percentages. The survey collects information in the last 28 days. We attempted a longitudinal study (not yet published) but of the 900 original patients only about 120 attended for a follow-up visit (2 months later) and only about 30 provided a response 2 months after their follow-up visit. While a lack of funding precluded ongoing involvement of a project Research Assistant to engage participants and conduct follow-ups, we note that we have captured patients throughout their cancer journey with patients having between 25 and 600+ days of treatment and follow-up.

---------------------------------------------------------------------

Comments on the Quality of English Language

6. As noted in the previous review, the paper would benefit from review by a copy-editor. The added text in tracked changes contains many errors and it is difficult to follow in many cases.

RESPONSE

Thank you for your comment. Consistent with the journal’s revision processes, we provided both the marked up and the clean copy of the manuscript for this reason. We would suggest that the clean copy would be easier to read and we corrected some minor spacing issues and removed some duplicate words in the clean copy. However, we will defer to the Editor as to which versions should be provided for peer review.

Reviewer 3 Report (Previous Reviewer 3)

Comments and Suggestions for Authors

The authors have revised the manuscript according to the review's comments. Thus, the manuscript can be accepted after proofreading.

Comments on the Quality of English Language

Proofreading is necessary.

Author Response

The authors have revised the manuscript according to the review's comments. Thus, the manuscript can be accepted after proofreading.

Comments on the Quality of English Language

Proofreading is necessary.

RESPONSE

We thank reviewer #3 for their acceptance and we have now conducted a comprehensive proofreading edit and corrected minor typographical and grammatical errors.

Reviewer 4 Report (New Reviewer)

Comments and Suggestions for Authors

The authors have looking at out of pocket costs and the effect on patients cancer treatments. It is a very interesting analysis which gives some pause on what patients go through. I think it is an important study to have out there. My one big comment is the available version has a bunch of track changes on it which makes it difficult to read. I think a clean version should be submitted for re-review.

1. I am not sure you should lump 4 provinces and call it Atlantic. As each province likely has different drug coverage I think it would be important to spell out which 2 centers were involved (where they from 1 province for example).

2. I think you have to be careful calling this a Canadian study. As mentioned the study only focused on BC, Alberta, Manitoba and Ontario which is only 40% of the provinces and no territories. I would choose something slightly modified.

3. There is mention that low economic pop was represented in the study. I think it should be mentioned that this population may also not be seeking medical treatment in the first place (there is a fair amount of data in regards to the types of patients who seek cancer treatment). This sounds like it solidifies that.

4. What about a comparison of age. I would assume the younger you are, the more likely your working to support your family, and the more burdensome OOPC would be. 

There are also several spelling and grammatical errors that will need to be changed. Example line 156, The word imputed I assumed should have been inputted? 

Comments on the Quality of English Language

see above

Author Response

The authors have looking at out of pocket costs and the effect on patients cancer treatments. It is a very interesting analysis which gives some pause on what patients go through. I think it is an important study to have out there. My one big comment is the available version has a bunch of track changes on it which makes it difficult to read. I think a clean version should be submitted for re-review.

RESPONSE
Thank you for this comment. Consistent with the journal’s revision processes, we submitted both a marked-up and clean copy of the manuscript to the journal for revision 1 and we will do so again for revision 2. However, we will defer to the Editor as to which versions should be provided for peer review.

  1. I am not sure you should lump 4 provinces and call it Atlantic. As each province likely has different drug coverage I think it would be important to spell out which 2 centers were involved (where they from 1 province for example).

RESPONSE

Thank you for this comment. The total number of patients across the 4 Atlantic provinces was 19. There were also enrollments from Quebec (19) and Saskatchewan (13), and a grand total of 51 across these 6 provinces. The breakdown by Atlantic province was (Nfld 4, NS 9, NB 4, PEI, 2). These numbers were all too small to provide statistically reliable information, hence were not analyzed and are excluded from the current study. We have changed this category to “Other provinces” since Atlantic is not accurate in this context. We have also changed the title of this paper to reflect that the analysis was based on 4 provinces in Canada.

-----------------------------------------------------------

  1. I think you have to be careful calling this a Canadian study. As mentioned the study only focused on BC, Alberta, Manitoba and Ontario which is only 40% of the provinces and no territories. I would choose something slightly modified.

RESPONSE

Thank you for this comment. All patients enrolled in this study were Canadian, hence it is a Canadian study.  Unfortunately, recruitment in some provinces was limited hence the analysis of all provinces was not possible (see previous comment). To clarify this for the reader, we have added an additional statement to the results section, and as mentioned above, we have changed the title of the paper.  It is also noteworthy that these 4 provinces represent 63.5% of the Canadian population (2021 census).

-----------------------------------------------------------

  1. There is mention that low economic pop was represented in the study. I think it should be mentioned that this population may also not be seeking medical treatment in the first place (there is a fair amount of data in regards to the types of patients who seek cancer treatment). This sounds like it solidifies that.

RESPONSE

Thank you for this comment.  I agree this is indeed an important consideration and will add this in the limitation section.

-----------------------------------------------------------

  1. What about a comparison of age. I would assume the younger you are, the more likely your working to support your family, and the more burdensome OOPC would be.

RESPONSE

Thank you for this question. We did run an analysis across age and did not find a strong linear relationship, however we did see a significant difference when comparing those under 65 to those over 65 and this was reported. This was justified as most provinces have a number of benefits that are covered for the 65 and over population. Additionally we have added a multiple linear regression which shows that increased age reduces patient costs and added this to methods, results and the discussion.

-----------------------------------------------------------

There are also several spelling and grammatical errors that will need to be changed. Example line 156, The word imputed I assumed should have been inputted? 

RESPONSE

Thank you for your question. The reviewer has misunderstood the word should be imputed as the cost of travel was calculated as a product of distance travelled in km and the Canadian Revenue Agency rate of reimbursement.

Reviewer 5 Report (New Reviewer)

Comments and Suggestions for Authors

Overall the study makes a solid contribution to understand financial toxicity of cancer care in Canada. A few areas could be improved methodologically and the policy insights are timely and have potential to inform health system improvement efforts around financial protections and coverage.

Specific commenters are listed below:

·      More detail could have been provided on survey development and validation. What was the response rate? How representative was the final sample?

·      The recruitment process could be described more clearly. What were the inclusion/exclusion criteria? Were certain groups over/under-represented? Were respondents similar to non-respondents?

·      Examining differences by cancer stage would have added useful context to understand disease severity. Without this data, interpreting expenditure differences is more difficult.

·      Provincial sample sizes were uneven, limiting comparisons. Larger samples from more provinces would permit more granular jurisdictional analyses.

·      Multiple regression modeling would have been helpful, enabling controlling for confounders like age, income, etc. in provincial analyses.

Author Response

Reviewer 6 Report (New Reviewer)

Comments and Suggestions for Authors

Your paper highlights some fascinating results on the provincial differences in OOPC for cancer patients and families. As a health researcher, this was particular relevant for me regarding the content. However, as a whole, I found this submission to be poorly written, difficult to follow, and even the rounds of revisions this has gone through with other reviewers seem to be rushed. The lack of attention to detail can be clearly seen in not even following the journal's guidelines for submission (e.g., insertion of images instead for tables, insertion of website links instead of in-text citations). I also found it hard to follow which statistical tests you were using as Chi-squared tests suddenly appear in the results but your methods section only stated ANOVAs and t-tests. Unfortunately, I must recommend a rejection but have left you with comments below if other reviewers wish to see it published or if you wish to pursue publication elsewhere.

1. General. The citation style and subheading formatting doesn't follow MDPI's guidelines. Please address this.
2. Line 63. Replace "seetings" with "settings"
3. Line 73. "onestudy" appears to be one word. Please change to "one study" if this is the case.
4. Line 90-94; 108-111. Since the in-text citations in MDPI journals are numbered, I would suggest using Roman numerals here. So, it would read "...conducted to (i) quantify financial....; (ii) identify differences in relation..." and so on. Otherwise, it looks a little strange at first glance. Feel free to disregard my comment if you think I'm being too picky.
5. Line 112-113. Please cite the www.cancer.ca website like all other references.
6. Abstract; Line 115. I would suggest writing out British Columbia instead of BC as this journal is for a global readership.
7. I liked your "Data sources" subsection as it was very detailed.
8. Line 165. Change "thesmall" to "the small"
9. General. I noticed there were many instances of reliance on parentheses throughout your paper (e.g. (midpoint of study recruitment), (a conservative estimate), (considered higher risk groups)). Whenever I find myself adding parentheses, it's a sign that sentence structure could be improved. Please revise those sentences as we want to be detailed yet still make it easy for the reader to go through your article. For example, one sentence could be rephrased to "...Canada Revenue Agency mileage rates during the midpoint of study recruitment in 2018." The CRA website where you drew this information from should also be cited here.
10. Line 190-192. What was the HiREB number? Please report here.
11. Tables 1-3. Did you insert images of your original tables? Please insert tables just as you would in a Word document and then reformat them accordingly to MDPI guidelines.
12. Tables 1-3. It could be beneficial to add a footnote to the tables indicating that all dollar amounts listed are in Canadian dollars.
13. Line 207. Change "typesPatients" to "types. Patients"
14. Methods, Lines 181-188. The Tukey HSD test is called "Tukey Honest(ly) Significant Difference" rather than "Statistical".
15. Methods, Lines 181-188. You state "...and the appropriate t-test was then applied" which I'm confused about. Do you mean the nonparametric equivalent of the t-test (i.e., Mann-Whitney U, Wilcoxon)? Please clarify it in the main text. 
16. Methods, Lines 181-188. I noticed you reported some Chi-squared tests in your results, but there was no indication of you intending to do so in your methods section. Perhaps also indicate wich variables you are applying Chi-squared tests to.
17. Line 278. "The in this analysis..."?
18. Line 328. Change "of$600" to "of $600"
19. Results. I found myself re-reading the results section to try to pick out what was going on. I would strongly suggest giving your results section more thought, and perhaps even tabulating the p-values from ANOVAs, t-tests, and Chi-squared tests. The ideas and findings are interesting.

Comments on the Quality of English Language

See above.

Author Response

a. Your paper highlights some fascinating results on the provincial differences in OOPC for cancer patients and families. As a health researcher, this was particular relevant for me regarding the content. However, as a whole, I found this submission to be poorly written, difficult to follow, and even the rounds of revisions this has gone through with other reviewers seem to be rushed. The lack of attention to detail can be clearly seen in not even following the journal's guidelines for submission (e.g., insertion of images instead for tables, insertion of website links instead of in-text citations).

RESPONSE

Thanks for your comments. We have double checked the guidelines and have reached out to the editorial team, suggesting they can make these adjustments in the proof reading stage.

-----------------------------------------------------------------

b. I also found it hard to follow which statistical tests you were using as Chi-squared tests suddenly appear in the results but your methods section only stated ANOVAs and t-tests. Unfortunately, I must recommend a rejection but have left you with comments below if other reviewers wish to see it published or if you wish to pursue publication elsewhere.

RESPONSE

Thank you for your comment.  We have updated the methods to reflect all test performed including chi-square tests and have added a multiply linear regression model.

-----------------------------------------------------------------

1. General. The citation style and subheading formatting doesn't follow MDPI's guidelines. Please address this.

RESPONSE

Thanks for this comment. As above we have now checked our formatting to ensure it aligns with MDPI’s guidelines and the editorial team, suggested they can make these adjustments in the proof reading stage.

-----------------------------------------------------------------
2. Line 63. Replace "seetings" with "settings"

RESPONSE

Thanks for catching this mistake. It has been corrected.

-----------------------------------------------------------------
3. Line 73. "onestudy" appears to be one word. Please change to "one study" if this is the case.

RESPONSE

Thanks for catching this mistake. It has also been corrected.

-----------------------------------------------------------------
4. Line 90-94; 108-111. Since the in-text citations in MDPI journals are numbered, I would suggest using Roman numerals here. So, it would read "...conducted to (i) quantify financial....; (ii) identify differences in relation..." and so on. Otherwise, it looks a little strange at first glance. Feel free to disregard my comment if you think I'm being too picky.

RESPONSE

Thanks for catching this formatting error.  The editorial team, suggested they can make these adjustments in the proof reading stage.

-----------------------------------------------------------------
5. Line 112-113. Please cite the www.cancer.ca website like all other references.

RESPONSE

Thanks for the guidance on this. The citation is there, not clear if you are asking me to provide the shorter version of the link.

-----------------------------------------------------------------

6. Abstract; Line 115. I would suggest writing out British Columbia instead of BC as this journal is for a global readership.

RESPONSE

Thanks for this feedback. We will make the adjustment.

-----------------------------------------------------------------
7. I liked your "Data sources" subsection as it was very detailed.

RESPONSE

Thanks for your feedback.

-----------------------------------------------------------------
8. Line 165. Change "thesmall" to "the small"

RESPONSE

Thank for this comment. The change has been made.

-----------------------------------------------------------------
9. General. I noticed there were many instances of reliance on parentheses throughout your paper (e.g. (midpoint of study recruitment), (a conservative estimate), (considered higher risk groups)). Whenever I find myself adding parentheses, it's a sign that sentence structure could be improved. Please revise those sentences as we want to be detailed yet still make it easy for the reader to go through your article. For example, one sentence could be rephrased to "...Canada Revenue Agency mileage rates during the midpoint of study recruitment in 2018." The CRA website where you drew this information from should also be cited here.

RESPONSE

Thanks for these comments. We have reviewed use of parentheses to minimize their use, and added a link to the CRA website.
-----------------------------------------------------------------

10. Line 190-192. What was the HiREB number? Please report here.

RESPONSE

Thanks for this comment. The HiREB number has been added.

-----------------------------------------------------------------
11. Tables 1-3. Did you insert images of your original tables? Please insert tables just as you would in a Word document and then reformat them accordingly to MDPI guidelines.

RESPONSE

Thanks for this comment. We understood from the statement that they needed to be PDF documents so we combined the three tables into one document as there was no option to submit three tables. We have sought guidance from the editorial team, and they suggested they can make these adjustments in the proof reading stage.

-----------------------------------------------------------------
12. Tables 1-3. It could be beneficial to add a footnote to the tables indicating that all dollar amounts listed are in Canadian dollars.

RESPONSE

Thanks for this comment. Another reviewer provided similar comments. We have now made adjustments to these tables accordingly.

-----------------------------------------------------------------
13. Line 207. Change "typesPatients" to "types. Patients"

RESPONSE

Thanks for this comment. These errors have been corrected.

-----------------------------------------------------------------
14. Methods, Lines 181-188. The Tukey HSD test is called "Tukey Honest(ly) Significant Difference" rather than "Statistical".

RESPONSE

Thanks for this comment. We have now consulted a statistician and revised this according to the most appropriate language for this test as suggested by the reviewer.

-----------------------------------------------------------------
15. Methods, Lines 181-188. You state "...and the appropriate t-test was then applied" which I'm confused about. Do you mean the nonparametric equivalent of the t-test (i.e., Mann-Whitney U, Wilcoxon)? Please clarify it in the main text. 

RESPONSE

Thanks for the opportunity to clarify this further. We were referring to how we specify variance inour t-tests according to th results of Levene’s test for equality of variances. In consultation with a statistician, we have revised this to reflect the appropriate wording of these tests.

-----------------------------------------------------------------
16. Methods, Lines 181-188. I noticed you reported some Chi-squared tests in your results, but there was no indication of you intending to do so in your methods section. Perhaps also indicate which variables you are applying Chi-squared tests to.

RESPONSE

Thanks for this comment. We have revised this wording in the methods and results sections in consultation with a statistician.

-----------------------------------------------------------------
17. Line 278. "The in this analysis..."?

RESPONSE

Thanks for this comment. This error has been corrected.

-----------------------------------------------------------------
18. Line 328. Change "of$600" to "of $600"

RESPONSE

Thanks for this comment.  It seems there are a number of instances of this in the Marked-up copy. We also sent a clean copy that it appears was not provided for peer review. We have no correct these mistakes throughout the each document. Although we would suggest the reviewer examine the clean version, we will defer to the Editor as to which versions should be provided for peer review.

-----------------------------------------------------------------
19. Results. I found myself re-reading the results section to try to pick out what was going on. I would strongly suggest giving your results section more thought, and perhaps even tabulating the p-values from ANOVAs, t-tests, and Chi-squared tests. The ideas and findings are interesting.

RESPONSE

Thanks for these comments. In consultation with a statistician, we have revised how we present these findings.

Round 2

Reviewer 2 Report (Previous Reviewer 2)

Comments and Suggestions for Authors

Thank you for your responses to my second round of comments on the paper, and I appreciate the edits to the paper made in response. In particular, I appreciate the effort to compare the respondents to the overall cancer populations via the registry, as I think that helps to strengthen the validity of the findings.

However, in spite of some of these improvements, I still feel the paper has some ways to go before it will be acceptable for publication. For example, while the addition of information about variation across provinces in covered services and OOP costs is helpful, the new information does not really help the reader understand what we might expect to see in particular with regards to cancer costs. In addition, the use of a citation and a requirement for the reader to go review the other paper being cited in order to understand the methods for this particular paper places too much burden on the reader; it would be much more helpful if the methods specifically discussed the questions being analyzed for this paper and how they were asked of respondents. I would also appreciate some justification for the use of only 28-day cost measures; I understand this is what was asked but it is not explicitly stated in the paper why this metric was specifically chosen.

Comments on the Quality of English Language

Unfortunately the journal editors are only showing me the marked-up version, instead of the clean version, for my review. Given this, I would still strongly recommend review and editing by a copy-editor before publication, as it is still at times difficult to read through without finding errors.

Author Response

Reviewer 4 Report (New Reviewer)

Comments and Suggestions for Authors

The authors have addressed my concerns. No other concerns at this time. Thank you.

Author Response

Thank you for your comments.

Reviewer 5 Report (New Reviewer)

Comments and Suggestions for Authors

Thanks to the authors for addressing my previous comments. I have no further comments.

Author Response

Thank you for your comments.

This manuscript is a resubmission of an earlier submission. The following is a list of the peer review reports and author responses from that submission.

Round 1

Reviewer 1 Report

Comments and Suggestions for Authors

Thank you very much for the opportunity to review your manuscript which surveyed patients with cancer and looked at direct and indirect costs to the patient/family. There was intra-provincial differences with Alberta having the highest cost. Due to the out of pocket costs, about 41% of patients required adjustment in the care needs with CAM and vitamins being the biggest ones.

Overall, this work furthers the knowledge regarding the economic impact of disease in patients living with cancer.

Comments on the manuscript:

1. Unfortunately the questionnaire did not capture postal code which would have potentially allowed for the accounting of differences in transportation between provinces (to account for the impact of a rural population). However, the survey captures the longest distances; could that data reported in aggragrate form?

2. To help explain the differences between provinces with income lost, including the median income per province (via Statistics Canada) might help provide clarity/context.

3. If there was information about the average price of parking between provinces, that might add some value (acknowledging that it may be difficult to find).

Author Response

Please see the attached comments to reviewers.

Reviewer 2 Report

Comments and Suggestions for Authors

Thank you for the opportunity to review this paper, which addresses an extremely important topic in health care and health insurance. The question of affordability of health care, especially for patients diagnosed with conditions requiring expensive treatment, is an important area for research that can contribute to meaningful policy changes to ensure that financial concerns are not a major impediment to treatment access.

Unfortunately, it is my opinion that this paper does not meet standards for scientific soundness so as to warrant publication. I feel that the survey strategy, as currently described in the methods section, introduces a high likelihood for bias in the respondent pool. Specifically, the different means by which the authors invited participants to respond to the survey across provinces likely contributed to the variation in number of responses across the provinces. In addition, it is difficult to believe that it was not feasible to estimate the population size of cancer patients who could reasonably respond to the survey, as the authors state; since the survey was targeted to patients receiving treatment at particular cancer centers, it seems highly likely the cancer centers could have provided patient counts to facilitate the calculation of the response rate. Without understanding the response rate for the survey it is difficult to gauge whether the findings are in fact generalizable to the broader cancer patient populations in Canada.

While the results are interesting and provide empirical support for stated concerns about financial toxicity, given my concerns with the survey sampling and fielding approaches noted above, I am not sure how best to interpret them in light of other research findings.

Finally, the Introduction presents some evidence that geographic variation exists in health care costs, which is well-known in the field, but the authors only cite Canadian sources and ignore the ample evidence from the United States. If the intent is to solely focus on Canada, this could be stated earlier on, but it did seem surprising to ignore the experience of a much bigger health care market just to the south.

Comments on the Quality of English Language

The paper would benefit substantially from editing to clarify and improve the text; this would go a long way to helping the reader follow the story. As currently written, there are a number of typos that make it difficult to follow the flow.

Author Response

Please see attached comments to reviewers

Reviewer 3 Report

Comments and Suggestions for Authors

This study examined provincial differences in patient spending for cancer care and reductions in household spending including decisions to forego care in Canada. In general, the study has clear study aims, a good research design and preliminary data analyses. Yet there are some issue need to be addressed to improve the quality of the study.

1. Have the authors calculated the minimum required sample size before conducting the survey? Have all the participants were randomized selected before conducting the survey?

2. Is it possible to do multi-variate regression analyses of the variables? Thus, the authors may have a more in-depth understanding of the factors which may influence the patient spending of healthcare and can make the results more robust to fulfill the study aim. If yes, the discussion part also requires a revision.  

3. Please use three-line table to describe the contents in the tables.

4. The reference style need to be unified, e.g., exhibiting DOI or not.

Comments on the Quality of English Language

The manuscript needs proofreading before acceptance. 

Author Response

Please see attached reply to reviewers.
